# Perceived usability of a store and forward telehealth platform for diagnosis and management of oral mucosal lesions: A cross-sectional study

**Michelle Roxo-Gonçalves**[1,2]*, **Marco Antonio Trevizani Martins**[1,3], **Manoela Domingues Martins**[1], **Carlos André Aita Schmitz**[2], **Rafael Gustavo Dal Moro**[2], **Otávio Pereira D'Avila**[2], **Dimitris Rucks Varvaki Rados**[2,4], **Roberto Nunes Umpierre**[2,5], **Marcelo Rodrigues Gonçalves**[2,5], **Vinicius Coelho Carrard**[1,2,3]

**1** Oral Pathology Department, School of Dentistry, Universidade Federal do Rio Grande do Sul, Porto Alegre, Rio Grande do Sul, Brazil, **2** TelessaudeRS-UFRGS, Universidade Federal do Rio Grande do Sul, Porto Alegre, Rio Grande do Sul, Brazil, **3** Department of Oral Medicine, Hospital de Clínicas de Porto Alegre (HCPA/UFRGS), Porto Alegre, Rio Grande do Sul, Brazil, **4** Department of Internal Medicine, Hospital de Clínicas de Porto Alegre (HCPA/UFRGS), Porto Alegre, Rio Grande do Sul, Brazil, **5** Department of Primary HealthCare, Hospital de Clínicas de Porto Alegre (HCPA/UFRGS), Porto Alegre, Rio Grande do Sul, Brazil

* michelleroxo@hotmail.com

**Data Availability Statement:** The data underlying this study are included as Supporting Information

## Abstract

EstomatoNet was created in the south of Brazil to provides specialist support over a web-based platform to primary care dentists for diagnosis of oral lesions. To evaluate the usability of EstomatoNet and to identify user perceptions regarding their expectations and difficulties with the system; and to compare the perceptions of regular users of the service to those of first-time users. Sixteen dentists were selected for the study: 8 were frequent users of EstomatoNet and 8 were residents who had never used the Platform. To assess usability, participants were required to request telediagnosis support for a fictional case provided by the research team. During the process of uploading the information and sending the request, users were asked to "think out loud," expressing their perceptions. The session was observed by an examiner with remote access to the user's screen (via Skype). After the simulation, users completed the System Usability Scale (SyUS), a validated tool with scores ranging from 0 to 100. The mean SyUS score assigned by frequent users was 84.7±6.6, vs. 82.2±9.3 for residents (satisfactory usability: score above 68). The difference between the groups was not statistically significant (Student t test, $P$ = .55). The residents group took longer (347.1±101.1s) to complete the task than frequent users (252.8±80.3s); however, the difference between the groups was not statistically significant (Student t test, $P$ = .06). In their subjective evaluation, users suggested the inclusion of a field to add further information on outcomes and resolution of the case and changes in the position of the "Send" button to improve workflow. The present results indicate satisfactory usability of EstomatoNet. The Platform seems to meet the needs of users regardless of how experienced they are; nevertheless, a few minor changes in some steps would improve the tool.

and have been uploaded to the Open Science Framework (DOI: 10.17605/OSF.IO/8ANG2).

**Funding:** This work supported by Fundação Instituto de Pesquisas Econômicas (FIPE) https://www.fipe.org.br/ and Hospital de Clinicas de Porto Alegre https://www.hcpa.edu.br/ with the Publication fees. No direct funding was received for this study. The cited funders had no role in study design, data collection, analysis, interpretation, or writing of the report. The first and last authors had full access to all data and had final responsibility for the decision to submit the manuscript for publication.

**Competing interests:** The authors have declared that no competing interests exist.

**Abbreviations:** PHC, primary health care; SyUS, System Usability Scale.

## Introduction

A fundamental principle of the Brazilian Unified Health System (SUS) is the constitutional guarantee of access to comprehensive, continuous health care, coordinated through networks, for all citizens. In the SUS, the primary care level is the entry point to the health care system [1–5] and should be capable of resolving most health conditions. Considering that assumption, oral health was introduced as an integral and inseparable component of the SUS in 2004, when the National Oral Health Policy was launched. The SUS oral health care network encompasses specialized dental care, with referrals coordinated by primary health care (PHC) services [6,7].

### Telediagnosis as a tool to enhance access to specialized care

Even though face-to-face encounters between health professionals and patients remain the gold standard for patient evaluation in all medical fields, telediagnosis is a useful alternative when the availability of specialized care is limited [8]. Taking this into account, TelessaúdeRS-UFRGS, a major university-based telehealth program in Brazil [9], has created EstomatoNet, a teledentistry service available free of charge to PHC physicians and dentists in the state of Rio Grande do Sul to enhance care, prevent unnecessary referrals to specialists, and decrease the time between referral and specialty consultations [10]. Established in 2015, EstomatoNet is a web-based platform developed by TelessaúdeRS-UFRGS, and currently handles 3.7 thousand monthly requests for teleconsultations and telediagnosis support.

However, despite the advances and contributions of telemedicine, many health professionals still resist this technology. This might be explained, at least in part, by the difficulties involved in learning how to use these systems [11].

### Usability in telehealth

Usability is defined as the ability to use a product for its intended purpose [12]. Assessment of usability allows difficulties to be identified and resolved, making sure that telehealth systems do in fact translate into benefits. Usability assessment encompasses features such as ease of learning, ease of retaining the know-how to repeat a task after some time, how fast tasks can be performed, low error rate, and subjective user satisfaction [13]. Thus, the primary aim of the present study was to evaluate the perceived usability of the EstomatoNet Platform. The secondary aim was to compare the perceptions of regular users of the service to those of health professionals who were using it for the first time.

## Methods

### Study design and recruitment of participants

This is a cross-sectional, observational study using a convenience sample. Among the 71 registered users of EstomatoNet, those 10 who used it most frequently were invited via e-mail to participate in the study. Those who agreed to participate signed an informed consent form and were included as the experienced user group. All were dentists working in the Brazilian Unified Health System. A control group of first-time users was selected among residents from the Integrated Residency Program in Oral Health at the UFRGS School of Dentistry. Residents were not familiar with the EstomatoNet Platform. The Universidade Federal do Rio Grande do Sul review board (GPPG 16–0440) approved this studie.

## Study procedures

Initially, individual Skype videoconferences were scheduled with each participant. During the videoconference, the participant was asked by an examiner to activate the "screen sharing" tool to allow observation and analysis of the interaction with the Platform.

After screen sharing was activated, the participant was guided to access and read the tutorial on how to request telediagnosis support through a link to EstomatoNet made available at the TelessaúdeRS-UFRGS portal [10]. This procedure lasted approximately 10 minutes.

## Outcomes—Evaluation of usability

**Observation and "think out loud" protocol.** After that, the examiner (a professional familiar with the EstomatoNet workflow) shared a simulated case via Skype, including clinical data and photographs, for the Platform test. The same case was presented to all participants. During this simulation, the participants were instructed to "think out loud", that is, to express their feelings and difficulties while performing the task. In usability research, this approach has been useful to identify problems in information systems [14,15, 16].

All simulations were recorded using a digital camera (Canon EOS Rebel T3, 12 megapixels, 18-55mm lens) to allow analysis of the participants' actions and expressions and to record the time taken to conclude the procedure. Each video from each participant was analyzed twice by the examiner, to mitigate the possibility of details being overlooked.

**System usability scale (SyUS).** Study outcomes included the duration of the interaction, the perceptions expressed by participants regarding the Platform, and usability of the system according to the System Usability Scale (SyUS) [17,18]. This validated questionnaire was composed of 10 items (Table 1) and is highly reliable [18–20] to evaluate usability in different systems [21,22]. The total SyUS score ranges from 0 to 100, with scores higher than 68 considered to be satisfactory [20]. The original language of the instrument was English; it was translated and cross-culturally validated for Portuguese [23].

## Sample size and power calculation

Considering that the present study used a convenience sample, its power was calculated according to method proposed by Borsci et al. [24]. To estimate the amount of problems reached by the sample in question, the following equation was used: $D = 1-(1-p)^n$, where p represents the raw p-value, n indicate the sample size and D, the percentage of problems reached by the sample. The weight of each problem was calculated as the sum of users that have detected it, while the count of the problems identified by each subject was used for calculating the raw p-value, and the means of calculating each individual's p-value.

**Table 1. System usability scale (SyUS).**

| |
|---|
| 1. I think that I would like to use this system frequently. |
| 2. I found the system unnecessarily complex. |
| 3. I thought the system was easy to use. |
| 4. I think that I would need the support of a technical person to be able to use this system. |
| 5. I found the various functions in this system were well integrated. |
| 6. I thought there was too much inconsistency in this system. |
| 7. I would imagine that most people would learn to use this system very quickly. |
| 8. I found the system very cumbersome to use. |
| 9. I felt very confident using the system. |
| 10. I needed to learn a lot of things before I could get going with this system. |

## Statistical analysis

Quantitative variables were expressed as means and standard deviations. Analyses were performed in PASW Statistics v. 18.

# Results

## Characteristics of the sample

Of the 10 invited regular EstomatoNet users, eight agreed to take part in the study (80% response rate). All invited residents agreed to participate.

The age of participants ranged from 22 to 46 years. Most participants (12 of 16) were female (Table 2). As expected, none of the residents had any previous contact with the Platform.

## Evaluation of perceived usability: Observation and "think out loud"

The platform consists of three screens. The first one collects the patient's personal information, the second has questions about the lesion (Fig 1), and the third screen provides a link to attach the photograph. The main difficulty pointed by participants (four dentist and five residents) were to fill the Individual Taxpayer number (ITN) (Fig 1, left screen). Other important question referred by one dentist and three residents was to fill the patient's National Health Registry (CNS) number (Fig 1). In addition, analysis of Screen 3 revealed two important difficulties: 1) the need to "save a draft" of the request before attaching a photograph and 2) after attaching a photograph, the need to return to Screens 1 or 2 (Fig 1) to find the "send" button. There were four difficulties pointed out by users. The positive and negative perceptions were reported during the simulated requests (Fig 2), whereas the frequencies of problems are depicted in the Tables 3 and 4.

## Evaluation of usability: System usability scale (SyUS)

Individual SyUS scores are shown in Table 5. Regarding previous use of the Platform, heterogeneous results were obtained for dentists, with the number of previous interactions (i.e., previous requests placed) ranging from 3 to 35 (14.1±10.2. The highest score was assigned by two users: the most experienced participant (35 previous requests) and one resident. Table 5 shows scores above 80 for both groups (dentists and residents).

**Table 2. Demographic characteristics of study participants.**

|  | Dentists | Residents |
|---|---|---|
| **Age (years)** |  |  |
| Mean | 35.3 | 25.5 |
| SD | 6.3 | 2.8 |
| Min-Max | 28–46 | 22–31 |
| **Sex** |  |  |
| Male | 3 | 1 |
| Female | 5 | 7 |
| **Time since graduation (years)** |  |  |
| Mean | 9.0 | 2.3 |
| SD | 5.7 | 1.1 |
| Min-Max | 2–17 | 1–4 |

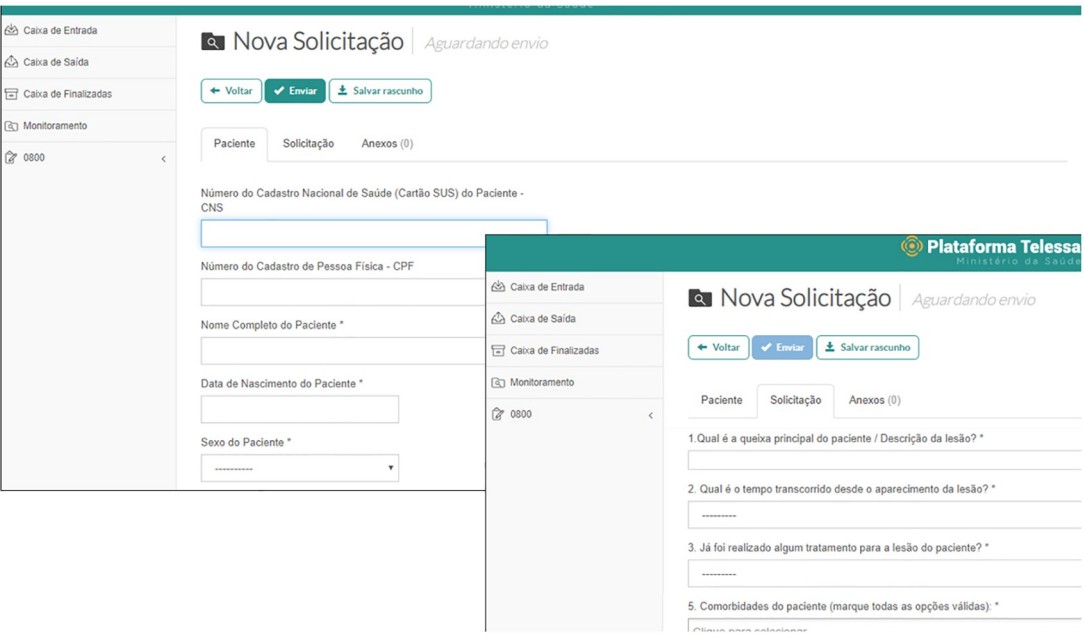

**Fig 1. Screens showing steps 1 and 2 required to request telehealth support.**

Tables 6 and 7 summarize the study findings. Residents required more time (347.1 s) to complete a request than dentists (252.8 s), who were familiar with the Platform workflow (Table 6). Data analysis showed an inverse correlation between SyUS score and time required to place the request (R = −0.54, *P* = .03, Pearson's correlation).

## Power of the sample

Following the method for sample power calculation the raw p-value for Residents and Dentists was 0.25 and 0.34, reaching, respectively 90% and 97% of the usability problems. The frequency of problems mentioned by each participant may be observed in the Tables 3 and 4.

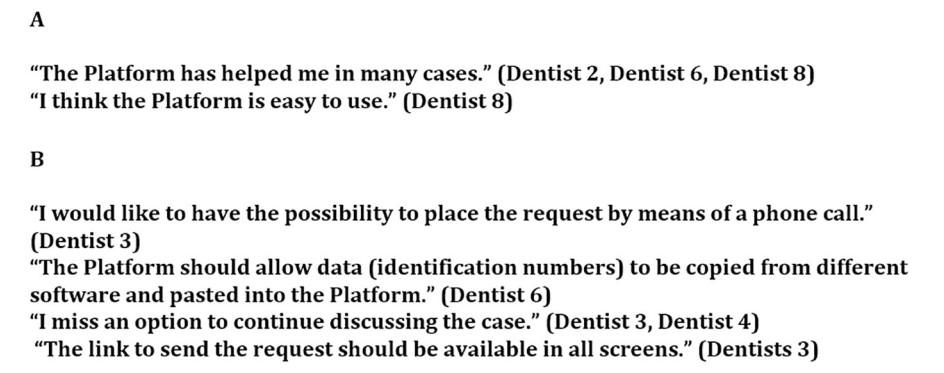

**A**

"The Platform has helped me in many cases." (Dentist 2, Dentist 6, Dentist 8)
"I think the Platform is easy to use." (Dentist 8)

**B**

"I would like to have the possibility to place the request by means of a phone call." (Dentist 3)
"The Platform should allow data (identification numbers) to be copied from different software and pasted into the Platform." (Dentist 6)
"I miss an option to continue discussing the case." (Dentist 3, Dentist 4)
 "The link to send the request should be available in all screens." (Dentists 3)

**Fig 2. Perceptions of residents or dentists regarding EstomatoNet.** (A) Positive comments. (B) Negative comments.

**Table 3. Frequency of difficulties referred by the residents during platform use.**

|         | Fill CNS | Fill ITN | Consent form | Workflow | Total (n) |
|---------|----------|----------|--------------|----------|-----------|
| Res 1   | 0        | 0        | 1            | 0        | 1         |
| Res 2   | 0        | 0        | 0            | 0        | 0         |
| Res 3   | 0        | 0        | 0            | 0        | 0         |
| Res 4   | 1        | 1        | 0            | 0        | 2         |
| Res 5   | 0        | 1        | 0            | 0        | 1         |
| Res 6   | 1        | 0        | 0            | 0        | 1         |
| Res 7   | 1        | 1        | 0            | 0        | 2         |
| Res 8   | 0        | 1        | 0            | 0        | 1         |
| **Total (n)** | 3  | 4        | 1            | 0        |           |

CNS—National Health Registry (CNS); ITN: Individual Taxpayer number

## Discussion

### Principal results

The creation of a platform to support PHC in Brazil was a significant effort to improve health care services in a large country with uneven professional training standards and different degrees of technological familiarity. In this scenario, assessing usability is crucial for improvement. The results show that EstomatoNet has satisfactory usability. Even though focal problems were detected, even first-time users of the tool did not experience major challenges.

It is known that 85% of dentists find it difficult to detect, diagnose, and treat oral lesions [25]. The establishment of a teledentistry platform is one way of addressing this need. The high usability score obtained by EstomatoNet shows that the platform has the potential to bridge this gap in an efficient manner [26]. Although the use of information technology in health services is promising, few high-quality studies have assessed it [27].

Usability assessed using the SyUS was fully satisfactory according to dentists and residents. The fact that the difference between the groups was not significant indicates that the Platform is user-friendly, easy to learn and to manipulate, even for inexperienced users. Statements such as "the Platform has helped me in many cases" or "I find the Platform easy to use", made by dentists who already use the Platform, confirm that it is well accepted. Finally, the statements regarding interest and self-confidence in using the system again support this interpretation.

**Table 4. Frequency of difficulties referred by the Dentists during platform use.**

|         | Fill CNS | Fill ITN | Consent form | Workflow | Total (n) |
|---------|----------|----------|--------------|----------|-----------|
| Dent 1  | 0        | 1        | 0            | 1        | 2         |
| Dent 2  | 1        | 1        | 1            | 1        | 4         |
| Dent 3  | 0        | 1        | 0            | 0        | 1         |
| Dent 4  | 0        | 0        | 0            | 0        | 0         |
| Dent 5  | 0        | 1        | 1            | 1        | 3         |
| Dent 6  | 0        | 1        | 0            | 0        | 1         |
| Dent 7  | 0        | 0        | 0            | 0        | 0         |
| Dent 8  | 0        | 0        | 0            | 0        | 0         |
| **Total (n)** | 1  | 5        | 2            | 3        |           |

CNS—National Health Registry (CNS); ITN: Individual Taxpayer number

**Table 5. Number of previous requests, SyUS score, and time spent for requests execution from participants.**

|  | Previous use (times) | SyUS | Time (s) |
|---|---|---|---|
| **Dentists** |  |  |  |
| Dent1 | 10 | 85,0 | 210 |
| Dent2 | 15 | 90,0 | 353 |
| Dent3 | 3 | 72,5 | 322 |
| Dent4 | 35 | 92,5 | 243 |
| Dent5 | 6 | 87,5 | 186 |
| Dent6 | 20 | 77,5 | 328 |
| Dent7 | 17 | 85,0 | 262 |
| Dent8 | 7 | 87,5 | 118 |
| **Residents** |  |  |  |
| Res1 | - | 72,5 | 510 |
| Res2 | - | 87,5 | 375 |
| Res3 | - | 72,5 | 413 |
| Res4 | - | 77,5 | 247 |
| Res5 | - | 100,0 | 222 |
| Res6 | - | 77,5 | 310 |
| Res7 | - | 87,5 | 270 |
| Res8 | - | 82,5 | 430 |

## Comparison with prior work

The high usability score obtained by the Platform becomes even more relevant if compared to the SyUS scores obtained by other tools. In a study by Ahn et al. [28], for example, in which five cardiopulmonary resuscitation training apps were assessed, only one app had a mean usability score above 80 ($81.17 \pm 19.01$). Lacerda et al. [29], who compared two cardiology tele-diagnosis interfaces, found a score below 80 for both (77.5 and 58.8). The differences among studies are probably related to the usability of the tools assessed therein.

Regarding the time required to complete the task (place a request), residents took longer than dentists. This suggests that the more one uses the system, the shorter the time to complete the task. A mean time below 6 minutes to perform the task (347.2s) seems acceptable to obtain the benefit associated with having support provided by a specialist who will help clarify the diagnosis and guide clinical decision-making. In addition, in many cases, specialist support may prevent referrals, reducing costs for the government and favoring professionals and patients.

**Table 6. SyUS score and time required to place a request according to study group.**

|  | Total | Dentists | Residents |
|---|---|---|---|
| **SyUS** |  |  |  |
| Mean |  | 84.7 | 82.2 |
| SD |  | 6.6 | 9.3 |
| Min-Max | 72.5–100.0 | 72.5–92.5 | 72.5–100.0 |
| **Time (s)** |  |  |  |
| Mean |  | 252.8 | 347.2 |
| SD |  | 80.3 | 101.1 |
| Min-Max | 118–510 | 118–353 | 222–510 |

**Table 7. Summary of answers obtained from the study participants (SyUS).**

| | Strongly disagree | Disagree | Neither disagree nor agree | Agree | Strongly agree |
|---|---|---|---|---|---|
| 1. I think that I would like to use this system frequently. | - | - | - | 6 | 10 |
| 2. I found the system unnecessarily complex. | 7 | 7 | 1 | 1 | |
| 3. I thought the system was easy to use. | | 1 | 1 | 8 | 6 |
| 4. I think that I would need the support of a technical person to be able to use this system. | 7 | 8 | 1 | - | - |
| 5. I found the various functions in this system were well integrated. | - | - | 2 | 10 | 4 |
| 6. I thought there was too much inconsistency in this system. | 8 | 7 | 1 | - | - |
| 7. I would imagine that most people would learn to use this system very quickly. | - | - | 1 | 7 | 8 |
| 8. I found the system very cumbersome to use. | 8 | 7 | - | | 1 |
| 9. I felt very confident using the system. | - | - | - | 12 | 4 |
| 10. I needed to learn a lot of things before I could get going with this system. | 8 | 14 | 2 | - | - |

The correlation test showed that the lower the time required to place the request, the higher the SyUS score. The fact that two unsatisfactory evaluations were made by the participants who took the longest to finalize the task further support this finding. Another factor influencing the SyUS score was experience with the Platform, since the professional with the highest number of previous requests (dentist 4) was the one who assigned the highest SyUS score.

The fact that some participants typed the name of the patient in the field meant for the National Health Registry number is justified by the name being, in general, the first information requested in web forms. The ability to copy/paste to the field and to view/access another window in the system could be considered improvements to the Platform.

In Screen 3, the steps of attaching a photograph and sending the request broke the workflow. Some simple adjustments would be enough to solve this problem. "Save draft" could also be available after the photograph is attached. The "Send" link could be transferred from screens 1 and 2 to screen 3. To avoid compromising task completion, this command could become automatically available as soon as the mandatory fields regarding the case were filled and the photographs attached.

The suggestion to add a field for further discussion of the case is also interesting. First, cases would be closed with information about outcomes, which would be useful for future reference; second, this adjustment would translate into an opportunity for continuity of care [30]. In other words, the specialist consultant could help the PHC dentist to evaluate the results of treatment in patients treated at the PHC level. with more effective follow-up along time.

## Limitations

The present study has limitations that need to be addressed. First, the small number of participants and the fact that all were recruited from the same state precludes extrapolation of the present findings to the country. Also, the number of interactions required to reach optimal usability was not assessed. Finally, the interactions assessed were simulations, and additional difficulties could perhaps arise during the placement of a request referring to a real case. All these issues could be the focus of future studies.

## Conclusions

The EstomatoNet Platform has satisfactory usability. Some focal problems associated with information fields should be addressed to improve the tool.

## Supporting information

**S1 Fig. Screens showing steps 1 and 2 required to request telehealth support.**
(PDF)

## Acknowledgments

The authors thanks to Rosely de Andrade Vargas and Claudia Buchweitz manuscript formatting final revision. The authors also thanks Erno Harzheim, Jéssica Rodriguez Strey, and Carlos Pilz for their contributions.

## Author Contributions

**Conceptualization:** Michelle Roxo-Gonçalves, Marco Antonio Trevizani Martins, Manoela Domingues Martins, Carlos André Aita Schmitz, Otávio Pereira D'Avila, Dimitris Rucks Varvaki Rados, Roberto Nunes Umpierre, Marcelo Rodrigues Gonçalves, Vinicius Coelho Carrard.

**Data curation:** Michelle Roxo-Gonçalves.

**Formal analysis:** Michelle Roxo-Gonçalves, Vinicius Coelho Carrard.

**Funding acquisition:** Michelle Roxo-Gonçalves, Roberto Nunes Umpierre, Marcelo Rodrigues Gonçalves, Vinicius Coelho Carrard.

**Investigation:** Michelle Roxo-Gonçalves.

**Methodology:** Michelle Roxo-Gonçalves, Carlos André Aita Schmitz, Vinicius Coelho Carrard.

**Project administration:** Michelle Roxo-Gonçalves, Vinicius Coelho Carrard.

**Resources:** Michelle Roxo-Gonçalves.

**Software:** Carlos André Aita Schmitz, Rafael Gustavo Dal Moro.

**Supervision:** Vinicius Coelho Carrard.

**Visualization:** Marco Antonio Trevizani Martins, Manoela Domingues Martins, Carlos André Aita Schmitz, Otávio Pereira D'Avila, Dimitris Rucks Varvaki Rados, Roberto Nunes Umpierre, Marcelo Rodrigues Gonçalves, Vinicius Coelho Carrard.

**Writing – original draft:** Michelle Roxo-Gonçalves, Vinicius Coelho Carrard.

**Writing – review & editing:** Michelle Roxo-Gonçalves, Marco Antonio Trevizani Martins, Manoela Domingues Martins, Carlos André Aita Schmitz, Rafael Gustavo Dal Moro, Otávio Pereira D'Avila, Dimitris Rucks Varvaki Rados, Roberto Nunes Umpierre, Marcelo Rodrigues Gonçalves, Vinicius Coelho Carrard.

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
