## [Decision Letter · Decision Letter 0]

30 Mar 2020

PONE-D-20-04286

Usability of a store and forward telehealth platform for diagnosis and management of oral mucosal lesions: a cross-sectional study

PLOS ONE

Dear Mrs Roxo-Gonçalves,

Thank you for submitting your manuscript to PLOS ONE. After careful consideration, we feel that it has merit but does not fully meet PLOS ONE’s publication criteria as it currently stands. Therefore, we invite you to submit a revised version of the manuscript that addresses the points raised during the review process.

We would appreciate receiving your revised manuscript by May 14 2020 11:59PM. To enhance the reproducibility of your results, we recommend that if applicable you deposit your laboratory protocols in protocols.io, where a protocol can be assigned its own identifier (DOI) such that it can be cited independently in the future. For instructions see: http://journals.plos.org/plosone/s/submission-guidelines#loc-laboratory-protocols

We look forward to receiving your revised manuscript.

Kind regards,

Simone Borsci, Ph.D.

Academic Editor

PLOS ONE

Journal Requirements:

Reviewers' comments:

Reviewer's Responses to Questions

**Comments to the Author**

1. Is the manuscript technically sound, and do the data support the conclusions?

Reviewer #1: Partly

Reviewer #2: Yes

2. Has the statistical analysis been performed appropriately and rigorously? 

Reviewer #1: No

Reviewer #2: Yes

3. Have the authors made all data underlying the findings in their manuscript fully available?

Reviewer #1: Yes

Reviewer #2: Yes

4. Is the manuscript presented in an intelligible fashion and written in standard English?

Reviewer #1: Yes

Reviewer #2: Yes

5. Review Comments to the Author

Reviewer #1: 1. The sample size calculation is based on a referenced manuscript that concluded 80% of a products usability problems can be detected with 4 or 5 subjects. It is not a true sample size calculation and further does depend on the likelihood of problem detection, the latter not estimated in this manuscript.

2. Also regarding the sample size estimation, it is not a sample size to detect either a difference or equivalence in the two outcomes assessed - a comparison between experienced users and inexperienced users in (1) SyUS scores and (2) time to complete the task. Therefore, there is no basis to assess the chance of an erroneous conclusion being reached based on the statistical tests applied to these outcomes. Given the current sample size rationale, these data would be better described descriptively.

3. It would appear that the response rate of 33% should read "non-response" rate based on the numbers provided by the authors, the authors should clarify.

Reviewer #2: The manuscript proposes a basic but solid assessment of satisfaction in use, which is not the entire assessment of usability, as per ISO standard 9241-11.

Overall the artcile is acceptable however, some raw statements should be removed or better specified.

First the title. As authors are not presenting measures of efficiency and effectiveness they can not claim that this study assessed usabilty but only statisfaction in use or perceived usability.

Second I know that the magic number of 5 users to discover 80% of the problems is still going around, but this is an old and complex story, and to make a long story short, NO five users are never enough - at max it could be a starting point. see for instance:

- Borsci, S., Macredie, R. D., Barnett, J., Martin, J., Kuljis, J., & Young, T. (2013). Reviewing and extending the five-user assumption: a grounded procedure for interaction evaluation. ACM Transactions on Computer-Human Interaction (TOCHI), 20(5), 1-23.

- Schmettow, M. (2012). Sample size in usability studies. Communications of the ACM, 55(4), 64-70.

I strongly suggest authors to remove references to the 5 users assumption.

6. PLOS authors have the option to publish the peer review history of their article (what does this mean?). If published, this will include your full peer review and any attached files.

Reviewer #1: No

Reviewer #2: No

---

## [Author Response · Author response to Decision Letter 0]

13 Apr 2020

Reviewer #1:

1. The sample size calculation is based on a referenced manuscript that concluded 80% of a products usability problems can be detected with 4 or 5 subjects. It is not a true sample size calculation and further does depend on the likelihood of problem detection, the latter not estimated in this manuscript.

Answer:

We follow the recommendation of Reviewer 2 regarding sample size in order to address this concern. The related changes are highlighted in yellow in the revised version of the manuscript.

2. Also regarding the sample size estimation, it is not a sample size to detect either a difference or equivalence in the two outcomes assessed - a comparison between experienced users and inexperienced users in (1) SyUS scores and (2) time to complete the task. Therefore, there is no basis to assess the chance of an erroneous conclusion being reached based on the statistical tests applied to these outcomes. Given the current sample size rationale, these data would be better described descriptively.

Answer:

Changes were made in order to follow the recommendations of both reviewers on this question. Furthermore, the data have now been presented descriptively (mean, SD and Min-Max).

3. It would appear that the response rate of 33% should read "non-response" rate based on the numbers provided by the authors, the authors should clarify.

Answer:

The recommend changes regarding sampling was performed and the abovementioned parameter was removed of the manuscript’s revised version.

Reviewer #2: 

1. The manuscript proposes a basic but solid assessment of satisfaction in use, which is not the entire assessment of usability, as per ISO standard 9241-11.

Overall the article is acceptable however, some raw statements should be removed or better specified.

First the title. As authors are not presenting measures of efficiency and effectiveness they may not claim that this study assessed usability but only satisfaction in use or perceived usability.

Answer:

We have changed the title to “Perceived usability of a store and forward telehealth platform for diagnosis and management of oral mucosal lesions: A cross-sectional study”.

2. I know that the magic number of 5 users to discover 80% of the problems is still going around, but this is an old and complex story, and to make a long story short, NO five users are never enough - at max it could be a starting point. see for instance:

- Borsci, S., Macredie, R. D., Barnett, J., Martin, J., Kuljis, J., & Young, T. (2013). Reviewing and extending the five-user assumption: a grounded procedure for interaction evaluation. ACM Transactions on Computer-Human Interaction (TOCHI), 20(5), 1-23.

- Schmettow, M. (2012). Sample size in usability studies. Communications of the ACM, 55(4), 64-70.

I strongly suggest authors to remove references to the 5 users assumption. 

Answer:

We would like to thank for the recommendation which will improve our manuscript. The mentioned references have been removed. Moreover, we performed the calculation proposed by Borsci et al. (2013) considering the findings obtained by the "think out load" evaluation. The four points of difficulty addressed by the participants were used for calculation, as referred by the suggested tool (Fig 5 - Borsci et al., 2013). The related changes are highlighted in yellow in the revised version of the manuscript.

---

## [Decision Letter · Decision Letter 1]

8 May 2020

Perceived usability of a store and forward telehealth platform for diagnosis and management of oral mucosal lesions: A cross-sectional study

PONE-D-20-04286R1

Dear Dr. Roxo-Gonçalves,

We are pleased to inform you that your manuscript has been judged scientifically suitable for publication and will be formally accepted for publication once it complies with all outstanding technical requirements.

With kind regards,

Simone Borsci, Ph.D.

Academic Editor

PLOS ONE

Additional Editor Comments (optional):

Reviewers' comments:

Reviewer's Responses to Questions

**Comments to the Author**

1. If the authors have adequately addressed your comments raised in a previous round of review and you feel that this manuscript is now acceptable for publication, you may indicate that here to bypass the “Comments to the Author” section, enter your conflict of interest statement in the “Confidential to Editor” section, and submit your "Accept" recommendation.

Reviewer #1: All comments have been addressed

Reviewer #2: (No Response)

2. Is the manuscript technically sound, and do the data support the conclusions?

Reviewer #1: (No Response)

Reviewer #2: Yes

3. Has the statistical analysis been performed appropriately and rigorously? 

Reviewer #1: (No Response)

Reviewer #2: Yes

4. Have the authors made all data underlying the findings in their manuscript fully available?

Reviewer #1: (No Response)

Reviewer #2: Yes

5. Is the manuscript presented in an intelligible fashion and written in standard English?

Reviewer #1: (No Response)

Reviewer #2: Yes

6. Review Comments to the Author

Reviewer #1: (No Response)

Reviewer #2: The paper is interesting and well written. I do not have suggestion to add, and I am happy to suggest to the editor publish this article as it is.

7. PLOS authors have the option to publish the peer review history of their article (what does this mean?). If published, this will include your full peer review and any attached files.

Reviewer #1: No

Reviewer #2: No

---

## [Editor Report · Acceptance letter]

28 May 2020

PONE-D-20-04286R1 

Perceived usability of a store and forward telehealth platform for diagnosis and management of oral mucosal lesions: A cross-sectional study 

Dear Dr. Roxo-Gonçalves:

I am pleased to inform you that your manuscript has been deemed suitable for publication in PLOS ONE. Congratulations! Your manuscript is now with our production department. 

With kind regards,

on behalf of

Dr. Simone Borsci 

Academic Editor

PLOS ONE